# Evolving epigenomics of immune cells at single-nucleus resolution in children en route to type 1 diabetes

Tomi Pastinen [1] ✉, Elin Grundberg [1], Todd Bradley [1], Jarno Honkanen[2], Warren A. Cheung [1], Arja Vuorela [2], Jeffrey J. Johnston [1], Byunggil Yoo[1], Santosh Khanal[1], Rebecca McLennan [1], Jorma Ilonen[3], Outi Vaarala[4], Jeffrey P. Krischer[5] & Mikael Knip [2,6,7] ✉

The appearance of diabetes-associated autoantibodies is the first detectable sign of the disease process leading to type 1 diabetes (T1D). Evidence suggests that T1D is a heterogenous disease, where the type of antibodies first formed implies subtypes. Here, we leverage longitudinal samples collected from 98 European TRIGR participants (49 children who subsequently presented with T1D, and 49 matched controls), and profile single-cell epigenomics at different time points of disease development. Quantitation of cell and nuclei populations, complemented by analysis of transcriptome and open-chromatin states, indicates robust, early, replicable monocyte lineage differences between cases and controls, suggesting the early emergence of heightened pro-inflammatory cytokine secretion among cases. The order of autoantibody emergence in cases shows variation across lymphoid and myeloid cells, potentially indicating divergence in the cellular immune response. The strong monocytic lineage representation in peripheral blood immune cells before seroconversion and the weaker differential coordination of these gene networks close to clinical diagnosis emphasize the importance of early life as a critical phase in T1D development.

Type 1 diabetes (T1D) is perceived as a chronic immune-mediated disease characterized by selective loss of insulin-producing β cells in the pancreatic islets of genetically susceptible individuals. Finnish children have globally the highest incidence of T1D, the risk has been estimated at 0.8% by the age of 15 years[1]. The average risk of T1D in children of European ancestry is about 0.25% by the age of 15 years[2], but it increases to 5% by the age of 15 years for children carrying risk associated HLA genotypes (DR3/DR4-DQB1*03:02)[3–5]. In addition to HLA there are over 100 other genetic loci described in the literature[6],

but collectively these genetic biomarkers do not reach high positive predictive value[7]. Symptomatic disease is preceded by an asymptomatic period of highly variable duration during which T1D-associated islet autoantibodies (IAA) appear in the peripheral circulation as markers of emerging β-cell autoimmunity[8,9]. In natural history studies, positivity for two or more autoantibodies signals a risk of approximately 70% for the development of clinical T1D over the subsequent 10 years[10]. Evolving data also suggest that T1D is not a homogenous disease but rather that there are different endotypes[11]. Such endotypes

[1]Genomic Medicine Center, Children's Mercy Kansas City and Children's Mercy Research Institute, Kansas City, MO, USA. [2]Research Program for Clinical and Molecular Metabolism, Faculty of Medicine, University of Helsinki, Helsinki, Finland. [3]Immunogenetics Laboratory, Institute of Biomedicine, University of Turku, Turku, Finland. [4]Orion Pharma, Espoo, Finland. [5]Health Informatics Institute, Morsani College of Medicine, University of South Florida, Tampa, FL, USA. [6]Center for Child Health Research, Tampere University Hospital, Tampere, Finland. [7]Turku Bioscience Centre, University of Turku and Åbo Akademi University, Turku, Finland. ✉e-mail: tpastinen@cmh.edu; mikael.knip@helsinki.fi

can be defined based on disease characteristics, such as which auto-antibody specificity appears first during the disease process[12,13], or the age at the manifestation of clinical T1D disease[14]. These observations required longitudinal studies of at-risk individuals, similar to The Trial to Reduce IDDM in the Genetically at Risk (TRIGR) study, including intense longitudinal follow-up starting at an early age (3 months onwards) coupled with rigorous biobanking of peripheral blood mononuclear cells (PBMC) along with other biospecimens[15]. Therefore, TRIGR provides a unique opportunity to observe immune-cell phenotypes in an at-risk population. TRIGR was a randomized clinical trial designed to assess whether it is possible to prevent β-cell autoimmunity and clinical T1D by weaning high-risk infants to an extensively hydrolyzed formula. The primary outcomes of the trial turned out to be negative[16,17].

Here, we extend these efforts, leveraging TRIGR and a time-resolved sample series of 98 participants (49 cases who progressed to clinical T1D and 49 age-matched controls), and profile single-cell epigenomics early, before the first signs of autoimmunity (autoantibody seroconversion), soon thereafter, and close to the manifestation of clinical T1D. The goal of our study is to illuminate early peripheral blood epigenomic changes prior to existing clinical biomarker positivity in matched at-risk patients with different disease outcomes. We perform population cell and nuclei analysis, including transcript and open-chromatin quantitation across immune cell lineages in both T1D cases and controls. Finally, we integrate the order of autoantibody emergence in the case group and showed variation in immune cells, potentially indicating cellular immune response divergence among T1D endotypes. We increase the cellular resolution of previous bulk-based analyses of T1D immune cells to understand divergence of early life immunity in patients developing autoimmunity and clinical disease. Our approach, more generally, with sample collection in pre-symptomatic at-risk population and application of single nuclei profiling, shows path for novel epigenomic biomarker development needed for future disease prevention.

## Results

### Longitudinal single-cell and single-nuclei profiling of peripheral blood cell types

To determine the immune cellular and regulatory landscape underlying T1D, we analyzed 98 individuals (Table 1, 49 progressing to T1D and 49 non-progressors) enrolled in the TRIGR study

each having peripheral blood mononuclear cells collected at three time points. Overall study design is depicted in Fig. 1a. The first time point in the cases was before seroconversion at the mean age of 1.6 years (range 0.3–6.1 years), the second time point soon after the seroconversion at the mean age of 3.0 years (range 0.8–8.2 years) and the

third time point close to the time of T1D diagnosis at the mean age 6.0 years (range 1.5–13.0 years). The time points in the controls corresponded to similar ages in the cases. Genetically unique samples ($N = 98$) were pooled prior to capture allowing us to minimize potential batch variation in downstream single nuclei or single cell assays. Three different captures were performed for each pool including (1) single cell RNA (scRNA), (2) single nuclei open chromatin (snATAC), and (3) multiomics snATAC and snRNA. A total of 187,995, 186,449, and 263,308 singleton (cells or nuclei) passed QC in scRNA, snRNA and snATAC assays, respectively, yielding on average 2278 independent single cells or nuclei measurements per sample and time point (Fig. 1b–d; Supplementary Data 1). A total of 307,744 case and 330,006 control cells or nuclei passed the QC, including a total of 85,628 looser GADA and 125,789 strict IAA cells or nuclei and total of 82,487 and 136,331 matching control cells or nuclei (summarized counts of individuals and cells by dataset, timepoint and condition are in Supplementary Data 2). The clustering (Fig. 1e–g; Supplementary Fig. 4a-b) of scRNA, snRNA, and snATAC followed by inference of major cell types revealed no statistically significant differences (after multiple testing adjustment) in the frequency of major cell types between cases and controls at any given time point (Supplementary Fig. 4c–f). However, strong calendar age-dependent trends were observed for proportional changes in B-cell and natural killer (NK)-cell subtypes (Supplementary Fig. 5).

### Linking open chromatin with gene expression at single nuclei resolution across immune cell types

The full snATAC dataset including 263,308 nuclei was used to call 99,779 peaks or regions with open chromatin across immune cell types. These open chromatin regions show 51% overlap with regions mapped by DNAseI hypersensitivity (DHS) in lymphoid and myeloid lineages (see methods) published by ENCODE[18]. We evaluated correlations of gene expression and open chromatin among the 186,449 nuclei up to 1 Mb distance from each transcript start site (TSS) (Fig. 2a). We note that as expected correlations show 9:1 overrepresentation of high access chromatin and high gene expression as opposed to negative correlations between chromatin accessibility and gene expression. Similarly, most genes have multiple open chromatin regions correlating to its expression (Fig. 2b).

In total, we identified 31,162 nominally significant ($P$ value < 0.05, |r| >0.05) correlations between open chromatin and gene expression. As expected, the highest density of open chromatin regions was observed proximal to TSS (20 kb flanking) of the associated gene with a density corresponding to 0.9 links per kb. Distal open chromatin and gene expression correlations (20–500 kb) were seen at a rate of 0.02 links per kb but even very distal associations (500 kb–1 Mb) were

**Table 1 | Characteristics of the study participants**

| | Cases (n = 49) | Controls (n = 49) |
|---|---|---|
| Sex, M/F | 31/18 | 34/15 |
| Age at end of follow-up, yr (range) | 6.58 (1.01–13.66) | 5.88 (0.76–13.01) |
| Age at firstsampling | 1.61 (0.25–6.03) | 1.68 (0.22–6.11) |
| Age at second sampling | 3.05 (0.75–8.18) | 3.25 (0.76–8.07) |
| Age at third sampling | 6.02 (1.53–12.99) | 6.27 (1.47–13.01) |
| HLA genotypes, n (%) *DQB1\*02/DQB1\*03:02 DQB1\*03:02/x DQA1\*05-DQB1\*02/y DQA1\*03-DQB1\*02/y* | 20 (40.8) 23 (46.1) 6 (12.3) | 9 (18.4) 15 (30.6) 19 (38.8) 6 (12.3) |
| Autoantibodies (AAB) during follow-up IAA, n (%) GADA, n (%) IA-2A, n (%) ZnT8A, n (%) ≥ 2 biochemical AAB, n (%) Number of biochemical AAB, median (range) IAA first, strict criterion, n (%) IAA first, looser criterion, n (%) GADA first, strict criterion, n (%) GADA first, looser criterion, n (%) | 44 (89.8) 43 (87.8) 40 (81.6) 31 (63.3) 46 (93.9) 1.47–13 3 (1–4) 21 (42.9) 38 (77.6) 8 (16.3) 11 (22.4) | |
| Age at initial seroconversion, yr (range) Duration of preclinical T1D, yr (range) | 2.26 (0.74–7.05) 4.20 (0.15-9.219) | |

x ≠ DQB1\*02, DQB1\*03:01, or DQB1\*06:02;

y ≠ DQA1\*02:01-DQB1\*02, DQB1\*03:01, DQB1\*06:02, or DQB1\*06:03

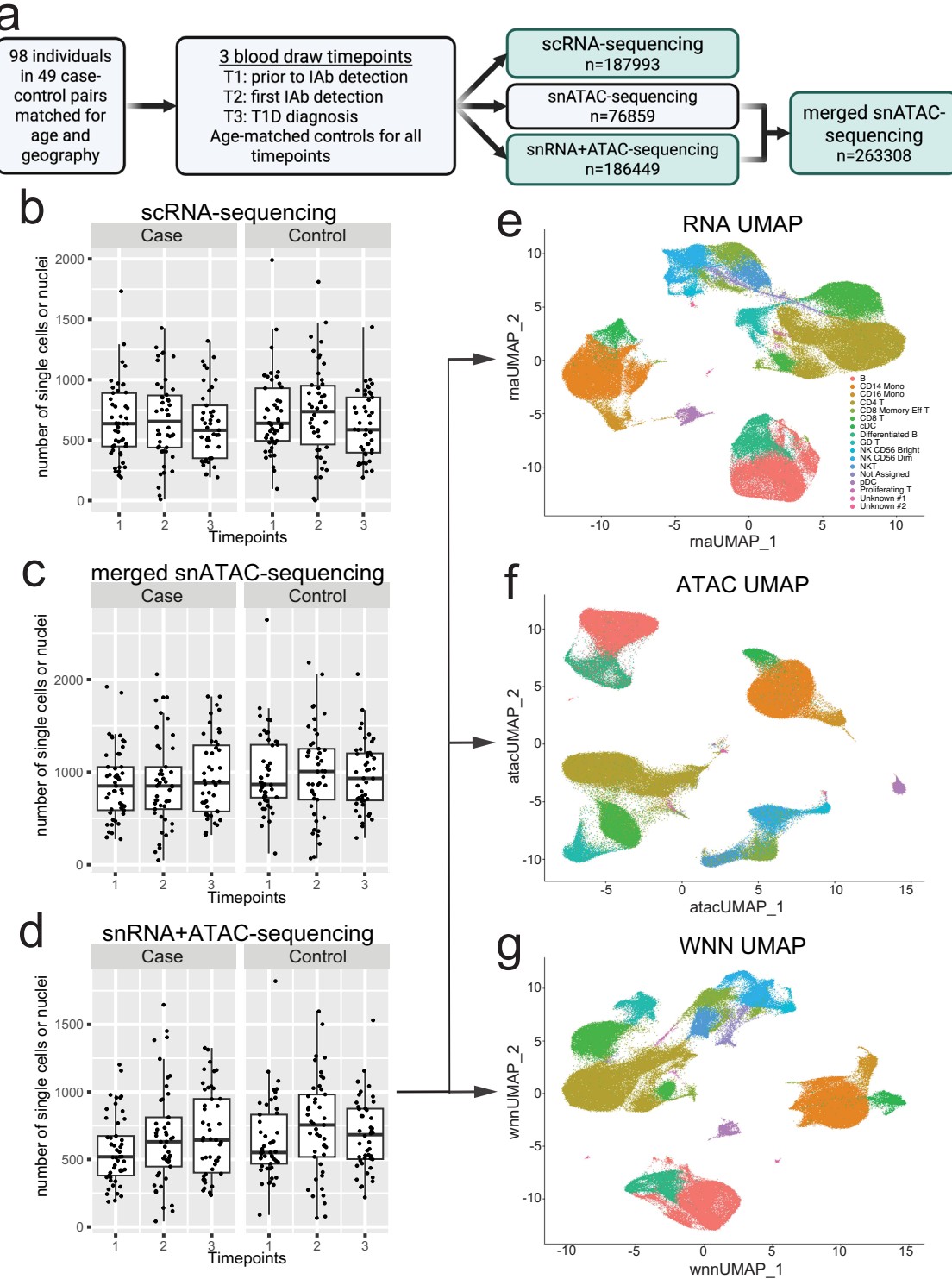

**Fig. 1 | Primary cell and nuclei data design and global distribution across full sample set. a** Overall study design and retained, quality controlled (QC), singleton samples from single nuclei (sn) and single-cell (sc) RNA (snRNA/scRNA) and open chromatin (snATAC) sequencing. Created with BioRender.com. **b**–**d** QC'd nuclei and single-cell counts per time point in cases and controls. Boxplots show median, with hinges at first and third quartiles and whiskers to the largest value no further than 1.5 * IQR from the hinge. N of samples (grouped by dataset, timepoint, and subtype) are summarized in Supp Data 2. Source data are provided as Supp Data 1. Major cell clusters from snMultiome nuclei in individual snRNA (**e**) and snATAC (**f**) or merged data (**g**) in Seurat/Signac analyses. T1, timepoint 1, T2, timepoint 2, T3, timepoint 3.

detected (0.01 links per kb). However, given the width of regulatory space explored the most abundant open chromatin and gene expression associations were seen at distal sites (Table 2). To address the issue of genome-wide multiple testing, we leveraged the independent

scRNA data in the same individuals but in independent capture, as well as independent snATAC from same individuals (unpaired scRNA and snATAC), to validate correlation of RNA and open chromatin mapped in snMultiome. With a strong correlation threshold (pearson |r | >0.65),

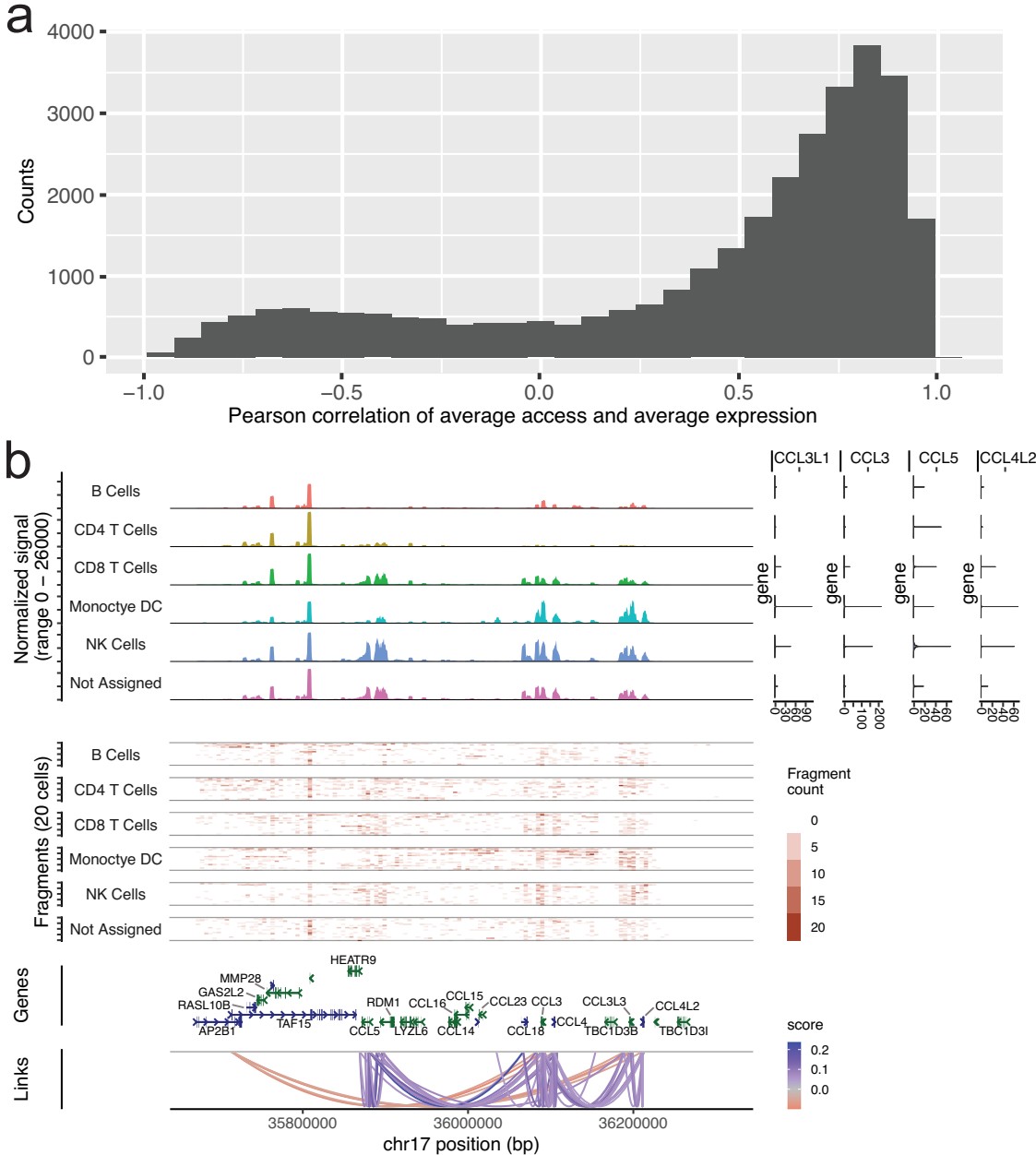

**Fig. 2 | Characteristics of open chromatin (snATAC) peaks as compared to nuclear (snRNA) -and cytosolic RNA expression in same individuals.** SnMultiome analyses by Signac reveals linked open-chromatin and nuclear RNA expression states in peripheral blood mononuclear cells from cases and controls at risk for T1D through early infancy to school-age years. **a** Initial set of 31,397 peaks–gene links (Table 2) from snMultiome was replicated in independent snATAC and scRNA analyses from same cohort of cases and controls. Over half of the nominally significant links showed concordant behavior in the replication test at absolute pearson r > 0.65. The distribution of replication pearson correlations depicted on the histogram show correlation distribution skewed to the right indicating predominantly positive relationships with chromatin openness and gene expression levels. Source data are provided in the Source Data file. **b** Example from chemokine locus in chr17 from snMultiome peak–gene link analyses showing open chromatin profiles across collapsed cell lineages (top left) together with linked gene expression levels (top right). Read depths for snATAC are shown as fragment count graphs (middle) and the significant peak/gene pairs called by Signac are shown in arcs (bottom).

**Table 2 | Multiome regulatory maps in PBMCs**

| Region in relation to gene TSS | snATAC Peaks (Macs2), snRNA genes, Links | Proportion of links \|r\| > 0.65 in scRNA |
|---|---|---|
| Promoter region (<5 kb) | 1865 links 1844 peaks 1460 genes | 64% 64% 65% |
| Proximal enhancer (5–20 kb) | 1936 links 1864 peaks 1261 genes | 67% 67% 69% |
| Distal enhancer (20–500 kb) | 17,368 links 10,534 peaks 3490 genes | 56% 64% 69% |
| Long range enhancer (0.5–1 Mb) | 10,218 links 6429 peaks 2692 genes | 47% 55% 63% |
| *Total* | 31,397 links 14,593 peaks 4213 genes | 54% 69% 71% |

altogether >50% of links were independently replicated showing same direction of effect (Table 2). These data provide core immunoregulome in the developing, circulating immune system extending to even distal regulatory sequences of common immune system genes. All open chromatin and gene expression correlations are listed in Supplementary Data 3.

## Differential multiomics immune signatures among T1D cases vs controls

Initial differential expression analyses indicated high proportion of associations among highly expressed genes in small cell populations observed in only few samples. To mitigate this outlier bias in small cell clusters, we utilized a mapping strategy collapsing related clusters to Monocyte, B-cell, CD4 T-cell, CD8 T-cell, and NK cell "parent lineages" for differential analyses between cases and controls. At a false discovery rate (FDR) of 10% across time points and cell lineages, we observed a total of 23,903 snRNA, 15,625 scRNA, and 5487 snATAC associations, respectively. Individual case−control associations feature associations show high variability (Fig. 3a). Consequently, we investigated the replication of cell lineage specific signals across the orthogonal data layers (time point, snRNA, scRNA, snATAC) for each individual molecular feature (Fig. 3b, c). Overall, original discovery significance predicts replication strongly in snRNA and scRNA and throughout data types, which are also correlated with the median read depths of the features and the number of replicated data layers (Fig. 3d). The highest discovery rate and largest number of replicated genes is observed in snRNA data, which also shows the highest read depth per feature (all features at significance qv <0.01 in case-control or subgroup analyses discussed below are listed in Supplementary Data 4–12). Across all data layers the level of replication is more than expected by chance. For example, in the case-control differential expression, we expect a significant number of genes observed in T1 to be replicated in T2, and similarly between T2 and T3. We observe non-random enrichment between observed T1/T2 and T2/T3 overlapping genes/peaks compared to the expected overlap even at lenient $P$ value thresholds, which grows stronger as the stringency of the significance is increased: at q < 0.5 we see a small enrichment (snRNA 2-2.6X, ATAC 1.6X-4X, scRNA 1.7-3.3X), but as we increase to q < 0.1 we observe a substantial enrichment (snRNA 4.8X-7.4X, ATAC 5.2X-507X, scRNA 3.6X-11X).

We also queried differences in age-dependent expression by a linear expression separately in cases and controls across cell lineages (Supplementary Data 13) to understand if nuclear epigenome traits developed differently among groups (q-value cut-off 0.1). The age-dependent snRNA traits that were shared in both case and control groups were substantially more robust than group specific traits showing median q-value of 0.001 (median r2 = 0.11) as compared to median q-value of 0.03 (median r2 = 0.06). Over 80% of top (1%:tile) age-dependent associated traits were shared in cases and controls underscoring that majority of expression trajectories are independent of case status.

## Early transcriptome differences in T1D cases vs controls linked to proinflammatory monocyte signatures

The primary snRNA case-control associations (at 10% FDR) were marginally ($\chi^2 = 10.2$, df=1, $P = 0.0014$) more abundant before (T1 = 8684 genes, 6% of tests) than soon after seroconversion (T2 = 8363 genes, 5.7% of tests). However, markedly fewer genes ($\chi^2 = 161$, $P < 0.000001$) were differentially expressed close to T1D diagnosis (T3 = 6856 genes, 4.7% of tests) as compared to T1. These associations were unequally distributed across cell lineages with monocytes showing greatest number of associations ($N = 6681$ genes across 3 time points), followed by B cells ($N = 5312$ genes), CD4 T cells ($N = 4557$ genes), CD8 T cells ($N = 4162$ genes) and NK cells ($N = 3192$ genes). While read depth per lineage was correlated with discovery rate (r = 0.72) the strongest contributor to this was low read depth in NK-cells (r = 0.44 without NK-

cells) and the high discovery rate in monocytes was observed at lower overall read depth (91%) compared to CD4 T-cells.

These observations including pronounced case-control associations in early time points as well as monocyte-lineage specificity were concordant in the orthogonal scRNA data sets. In fact, the primary scRNA case-control associations (at 10% FDR) show more pronounced differentiation (T1 vs. T2, $\chi^2 = 154$, $P < 0.000001$; T1 vs. T3 $\chi^2 = 428$, $P < 0.000001$) between cases and controls at the first time point (T1 = 6302 genes, 5.5% of tests) before seroconversion as compared to soon after seroconversion or close to clinical T1D (T2 = 5070 genes, 4.4% of tests; T3 = 4253 genes, 3.7% of tests). For the lineage-specific associations in the scRNA dataset, 5098 genes across 3 time points were identified for monocytes, 4720 genes for CD4 T cells, 2374 genes for B cells, 1823 for CD8 T cells and 1704 genes for NK cells. We then focused the analysis to pursue GeneSet pathway, and gene ontology enrichments analyses among differential expressed genes replicated in at least 2 data layers (Fig. 4). Greatest number of enriched gene networks were observed in monocytes, and a subset of these enrichments were replicated in B -and NK-cells. CD4 T cells show less coordination of genes throughout the progression of autoimmunity preceding T1D (Fig. 4a). We also performed upstream regulator analyses of genes using TRRUST (Transcriptional Regulatory Relationships Unraveled by Sentence Based Text Mining)[19] and found monocytes with *NFKB1*, *STAT1*, *RELA*, *SP1* and *IRF1* signatures particularly upregulated (early) whereas B-cells demonstrate MHC regulators (*RFX*, *RFXANK*, *CIITA*, *RFX5*) enriched for in controls (Fig. 4b). The MHC regulator difference in across all timepoints is reflected as relatively higher B-cell expression of HLA Class I (*HLA-A*, *HLA-B* and *HLA-C*) among cases but upregulation of HLA Class II (*HLA-DQA1*, *HLA-DQB1*, *HLA-DRB1*) in controls.

## Open chromatin analysis confirms early monocyte-specific regulation in T1D cases vs controls

Similar to primary transcriptome analyses in case versus control the open chromatin signal robustness as measured by replication (either via linked gene expression or independent time point) were strongly dependent on read depth (Fig. 3d, right panel) but overall discovery significance was narrower reflecting lower power for primary chromatin associations (Fig. 3b, right panel). In total, 22.7% of the 5487 unique peak / cell type/ time point associations demonstrated evidence for primary case-control difference in chromatin openness with replication in at least one data layer.

We looked for evidence of regulators of chromatin openness among associated peaks within each cell type. Prevalence of transcription factor (TF) binding sites were extracted separately for peaks showing increased access in cases vs. controls and vice-a-versa using TFmotifView[20]. These analyses can reveal directional impact of certain TF networks. Consistent with TRUSST analyses of gene regulators from scRNA/snRNA gene expression (Fig. 4b) the strongest evidence for coordinated differences were observed for monocytes and *IRF1*-motif (IRF1.MA0050.2) showing a fourfold increase in occurrence among case enriched open chromatin ($P = 6.22E{-}19$) (Fig. 4c), whereas among T cells numerous TFs show relative overrepresentation among cases and B cells or NK cells yielded no significant differences of directional TF binding. The pathway analyses based on differentially regulated (case versus control) peak – gene links (Fig. 2) for chromatin peaks per cell lineage show monocyte predominance in pathway enrichments similar to scRNA/snRNA analyses with the early time point in CD8 T-cells and B-cells clustering together with monocytes (Fig. 4d). We also looked for regionally clustered open chromatin signal differences between cases and controls to explore potential chromatin domain-wide differences in regulation. We observed 1727 adjacent chromatin associations across time points (210 expected by chance) and noted many regions showing dense clustering (Supplementary Fig. 7). These may reflect higher order chromatin domains acting differently among cases and controls.

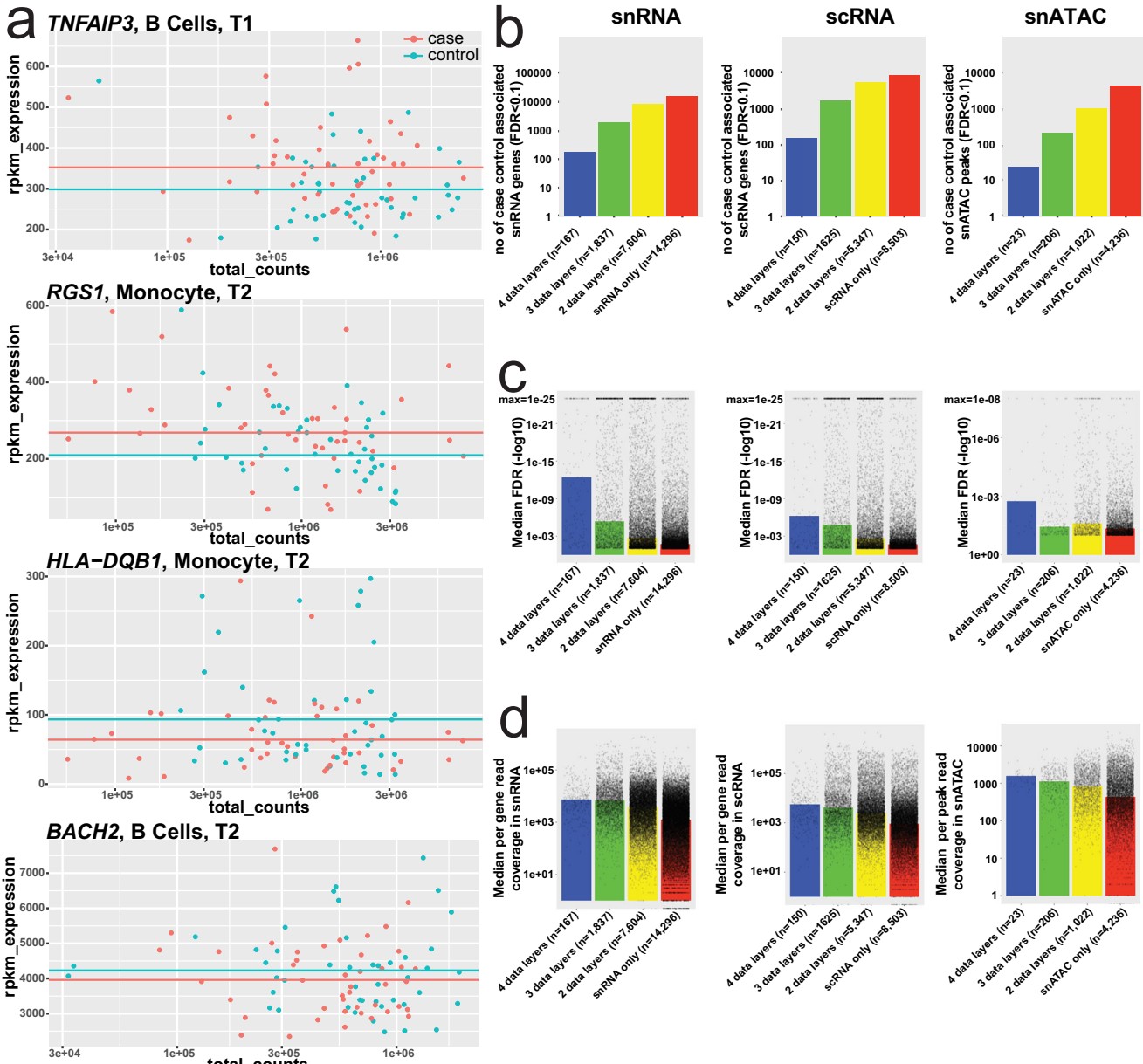

**Fig. 3 | Differential analyses of single cell or nuclei features in cases versus controls and their orthogonal reproducibility.** Source data for a-d are provided in the Source Data file. **a** Four examples of normalized (Reads Per Kilobase per Million mapped reads, rpkm; y-axis) per gene and total reads per individual (x-axis) used in statistical association to determine case (red dots) - control (green dots) differences in expression. Top graph shows *TNFAIP3* expression in B-cells at time-point 1 (T1), which is on average significantly ($p = 1.1e-36$) higher in cases (mean case rpkm shown as the red line) as compared to controls (green line). *RGS1* expression (second from top) is also significantly ($p = 5.8e-70$) higher in cases at timepoint 2 in monocytes. Whereas *HLA-DQB1* shows lower expression in case monocytes at timepoint 2 ($p = 9.5e-47$) and similarly *BACH2* (bottom) is relatively under expressed in B-cells at the same time point (two-sided Fisher's Exact test $p$-values from Supp Data 5). **b–d** Taking advantage of the study design with layers of data and multiple time points we queried the orthogonal data layers (time point, snRNA, scRNA, snATAC) for independent replication of each case-control

association. **b**, **c** The highest rate of replication is observed for scRNA, 45.6% seen it at least 2 layers of data, whereas 40.2% of more abundant snRNA signals are replicated. Much of the higher replication rate in scRNA is explained by consistent signals from ribosomal RNAs representing 6.0% of all replicated data points, whereas in snRNA only 1.4% of 2-layer positive case – control differences are mapping to ribosomal transcripts. This reflects the cytosolic scRNA overall molecular composition, with high proportion of ribosomal RNA reads, largely absent in snRNA (Supplementary Fig. 6). FDR corrected Fisher-test case-control $p$ values were used (Supp Data 4-6)−snATAC FDR $p$-values were plotted with a min of 1e-8, scRNA and snRNA FDR $p$-values were plotted with a min of 1e−25. **d** Open chromatin features, have lower discovery with less read coverage per snATAC peaks. Overall the median feature by group (case/control) read depths shown in the graphs panel correlate with replication level (r = 0.79 for ATAC, r = 0.82 for snRNA and r = 0.99 for scRNA).

## T1D genetic loci and single nuclei-epigenomes

We next sought to analyze differences in snRNA expression and open chromatin among cases and controls through the lens of genetic risk using loci from genome-wide mapping of T1D risk[6], including each of the 144 independent loci associated at FDR < 0.01. We associated each

index single nucleotide variant (SNV) for peak genotype with open chromatin or gene expression in the five cell lineages (CD4T, CD8T, NK, MONOCYTE, B CELL) at each time point (T1/T2/T3) using Spearman rank test and adjusted for multiple testing. We identified 244 associations (51 gene expression-SNV and 193 open chromatin-SNV

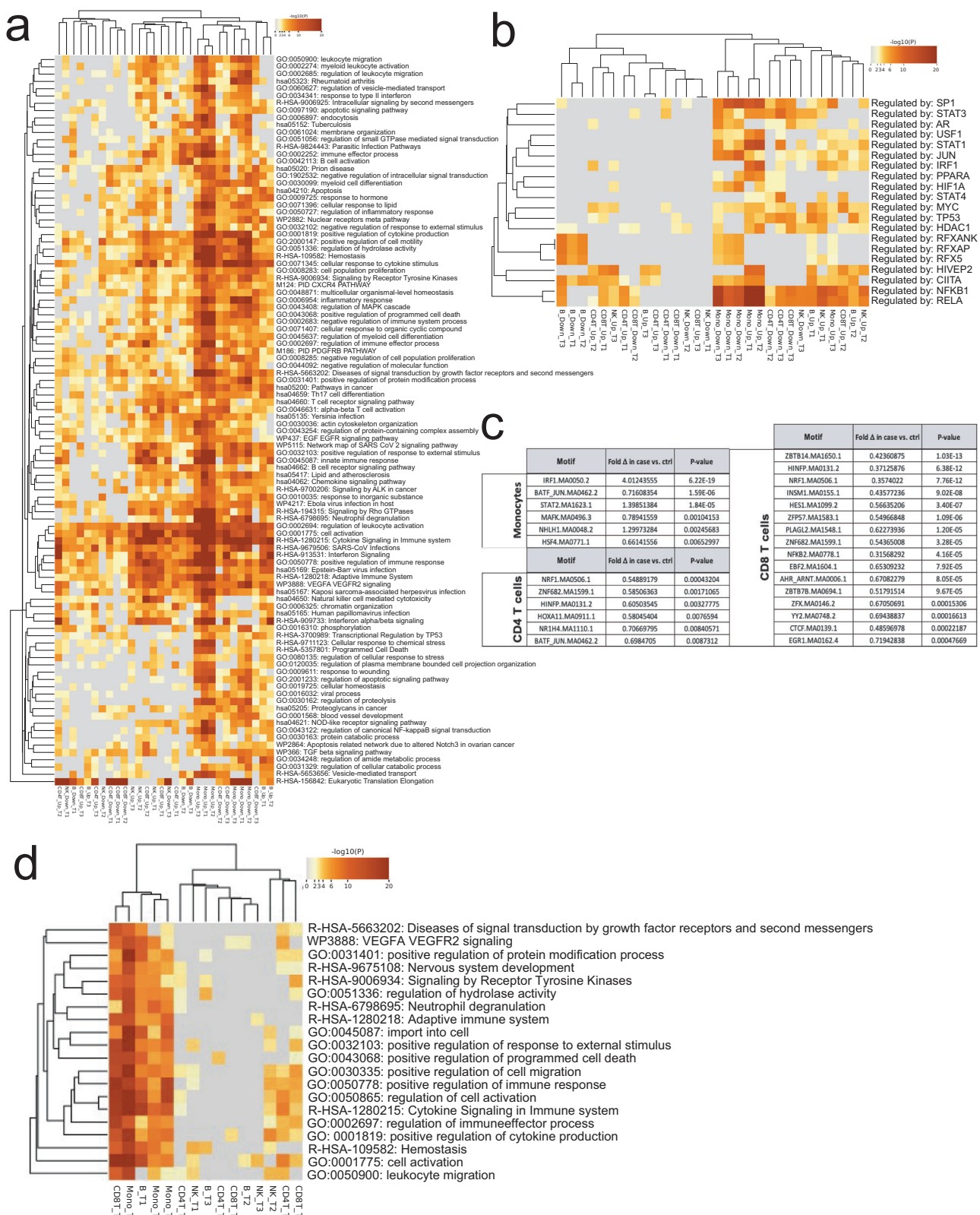

associations within 1 Mb of SNV, respectively) at 57 T1D loci (Supplementary Data 14). We examined these genetically linked loci in our case versuscontrol differential gene expression/open chromatin data and observed that 11/57 loci (19%) also showed significant differences between cases and controls. However, the rate of case versus control differences was not overall enriched among T1D SNV linked loci as compared to snRNA traits overall. Separately, we queried snRNA

expression differences from cases and controls for each of the named candidate genes at 144 loci[6], i.e., irrespective if a significant snRNA QTL was detected in our data. Among these, 94 (66%) were observed at least in one cell lineage/time point to be differentially expressed (10% FDR) in snRNA expression (Supplementary Data 15), where most genes appeared on multiple occasions (total n of case-control differences = 412). Among the top 50 most significant differences in disease genes

**Fig. 4 | Enriched gene networks and transcription factor binding site motif among differentially expressed genes and peaks. Proinflammatory monocyte signatures early monocyte-specific regulation in T1D cases vs controls. a** Input to enrichment analyses by Metascape (hypergeometric test $p$ values with Benhamini-Hochberg correction) were genes represented by snRNA/scRNA features and replicated in at least one data layer including a total of 2470 monocyte genes showing relative upregulation in cases vs. controls (Mono_Up) across 3 time points and 1975 relatively downregulated genes (Mono_Down), 1408 up -and 1282 down-regulated genes in CD4 T-cells (CD4T_Up/CD4T_Down) as well as 1281/1238 B_Up/B_Down, 857/751 CD8T_Up/CD8T_Down and 565/553 NK_Up/NK_Down genes.

**b** Upstream regulator analyses of genes using TRRUST (Fisher's Exact test with Benjamini-Hochberg correction). **c** Using 2-by-2 tests (two-sided Fisher's exact test) we determined significant differences in representation of individual motifs from TF motif in cases vs. controls and required nominal $P < 0.01$ for difference between open and closed chromatin peaks. **d** Differentially regulated gene networks between cases and controls revealed by gene – peak link analyses (Fig. 2) similarly highlight predominant strong predominance of early monocyte lineage responses as key differentiator. Generated by Metascape (hypergeometric test $p$ values with Benhamini-Hochberg correction).

between cases and controls (Table 3) we note relative upregulation of *NFKB1* and *TNFAIP3* across several cell lineages and time points among cases. In contrast, cases show predominant downregulation of *HLA-DQB1* and *BACH2*. Similarly, regions with genetic loci queried for differential chromatin activity between cases and controls revealed 155 independent peaks within 100 kb of index variant including 82 loci (Supplementary Data 16). The chromatin signals are significantly more common (42% excess, $\chi^2 = 11$, $P < 0.001$) at time point 1 in genomic intervals with T1D SNV as compared to genome wide differential peak accessibility between cases and controls, but as noted above the vast majority of these peaks do not reach significant genetic association with the T1D lead SNV.

### T1D endotypes and replicated snRNA traits

To gain insight into potential heterogeneity of single nuclei signals among cases we queried whether the order of appearance (IAA first strict criterion $n = 21$ / GAD first looser criterion $n = 11$) of autoantibodies as an endophenotype could explain any of the overall case−control patterns observed. We matched each subphenotype to its paired control for analyses to ensure age-related factors would not interfere with the analyses and compared the subgroup case against its control at each cell. We focused on features replicated in full case−control analyses (Fig. 3) given the smaller sample subpopulations and contrasted features that showed association in one endotype predominantly (Supplementary Fig. 8). We required a significant (qv<0.1), replicated (replication criteria across data layers as above) snRNA expression difference overlapped and matched the pattern seen in tests between cases and controls. On average cell lineage specific up -or downregulation in cases versus controls showed overlap of 25% with either "IAA first" or "GAD first" signal. However, in early time points (T1/T2) for monocytes the patterns in "GAD first" subgroup of patients were overlapping larger fraction of genes in same direction as seen in all cases and CD4 T-cell first timepoint showed larger overlap with "IAA first" subgroup (Fig. 5a). Consequently, the most prominent differences in all cases versus controls in pathway analyses (Fig. 4) seen for early monocyte overexpressed genes were recapitulated in "GAD first" subgroup data (Fig. 5b), whereas the "IAA first" subgroup shows enrichment for adaptive immune system networks in CD4 T-cells (Fig. 5c) before IAA-appearance (Timepoint 1).

### Discussion

Exploration of molecular features of human complex disease by genomics is rapidly expanding beyond DNA sequences (genotyping) alone to layers of gene regulation and gene expression[21]. Single-cell (or nuclei) experiments are most recent additions to "multiomics" tools, offering to reduce confounding impact of primary tissue heterogeneity. However, all multiomics approaches are limited by access to disease relevant tissues and generally by measuring tissue response to chronic disease state. The latter scenario, as demonstrated in a population single-cell study for systemic lupus[22], can identify known molecular differences in cases versus controls, but they are unable to provide prognostic or etiologic biomarkers.

The precise drivers of T1D remain unknown, but islet autoimmunity often occurs months to years before clinical disease onset.

Previous studies after the appearance of islet autoimmunity or T1D have identified differences in immune cells, including expansions of islet-reactive T cells, between individuals with islet autoimmunity and healthy controls[23]. Moreover, the targets of islet autoantibodies as well as the number of epitopes targeted are associated with differential increased T1D disease risk. One challenge of studying T1D in children is separating the impact that age has on the developing immune system from true changes in children with islet autoimmunity[24]. Here, we benefit from longitudinal sampling of a cohort of children at risk for T1D to pursue the first single cell multiomics case−control analyses months to years before the first clinical biomarkers appear or overt disease is present. The observation of differential interferon and cytokine signaling early before autoantibody formation and during seroconversion yields support from earlier gene expression studies in smaller, cross-sectional samples[25]. In contrast the genetically associated loci for T1D risk showed only modest enrichment for the signal in early time points. Many enriched pathways such as interferon response and proinflammatory *NFKB1* targets seen in monocytes of cases could be reactions to exogenous trigger since they are not known to be in the causal pathway for autoimmunity in type 1 diabetes. Albeit, rare variation in these regulators have been reported in autoimmune polyendocrinopathy[26].

Interestingly, the strong monocytic lineage representation in this response in peripheral blood and the relatively weaker differential coordination of these gene networks close to clinical diagnosis are an observation with important implications for follow-up studies. The endophenotype subgroup analyses suggests that this phenomenon is driven more prominently in patients with GAD antibodies appearing first. A recent pancreatic single-cell study of a rat model of autoimmune diabetes at pre-diabetic stage highlighted increased monocyte/macrophage chemokine and interferon response at onset of autoimmunity[27].

Despite measuring millions of cells and nuclei in the experiment and recovering 670,000 high quality, singleton nuclei and cells for analyses, the data for individual data layers suggests moderate discovery power correlated with reads per measured element and most prominently affecting single nuclei ATAC-seq data. However, the multiomic approach and primary analyses based on signals replicated in independent data types and/or second time point we present here was deployed to ensure robustness of data. The gene network enrichments point towards strong activation of innate immunity, interferon signaling, chemo/cytokine signaling, immune cell activation and inflammation biased to early (T1 and T2) time points and stronger in genes relatively upregulated in cases. A previous study performing longitudinal whole-blood transcriptome analysis of selected TEDDY participants before the onset of T1D also identified an innate immune cell gene signature that was associated with the individual T1D risk[28]. In that study an age-dependent NK cell gene signature correlated with progression to T1D. We had lower power for NK discovery and in TEDDY the majority of at-risk patients did not have familial T1D used as inclusion criteria for cases in TRIGR.

Some previously highlighted genetic risk factors (e.g., *BACH2, TNFAIP3*)[29,30] that are also quantitative traits were differentially expressed among cases and controls. However, most case/control trait

**Table 3 | Top 50 most significant differences in disease genes between cases and controls (FDR *p* values from Fisher's exact tests—see Supplementary Data 5)**

| Gene | Time point | Cell Lineage | Effect (Red=upregulated in cases; green=upregulated in controls) | FDR |
|---|---|---|---|---|
| *AFF3* | 1 | B Cells | 0.881882 | 3.788E−108 |
| *AFF3* | 2 | B Cells | 1.07308 | 8.6754E−24 |
| *AFF3* | 1 | CD4 T Cells | 0.850692 | 7.9557E−22 |
| *AHI1* | 1 | CD4 T Cells | 1.23272 | 4.816E−27 |
| *BACH2* | 2 | CD8 T Cells | 1.12515 | 1.455E−129 |
| *BACH2* | 3 | B Cells | 1.09819 | 4.742E−94 |
| *BACH2* | 1 | B Cells | 0.939154 | 2.5752E−77 |
| *BACH2* | 3 | CD4 T Cells | 1.07144 | 4.0642E−76 |
| *BACH2* | 2 | B Cells | 1.06456 | 9.8331E−55 |
| *BACH2* | 2 | CD4 T Cells | 1.04161 | 4.9935E−30 |
| *CD226* | 2 | Monocyte | 1.32033 | 4.1167E−26 |
| *CD6* | 1 | CD4 T Cells | 1.09239 | 1.7086E−20 |
| *CD69* | 1 | B Cells | 0.881482 | 1.572E−62 |
| *CD69* | 3 | NK Cells | 1.12464 | 2.7492E−40 |
| *CD69* | 1 | NK Cells | 0.88699 | 1.5986E−21 |
| *DENND1B* | 1 | Monocyte | 1.14323 | 9.693E−37 |
| *DENND1B* | 1 | B Cells | 1.17661 | 4.2708E−24 |
| *FYN* | 1 | NK Cells | 1.09758 | 1.1124E−34 |
| *FYN* | 1 | CD4 T Cells | 1.0491 | 7.8605E−29 |
| *HLA-DQB1* | 2 | Monocyte | 1.42182 | 3.0407E−44 |
| *HLA-DQB1* | 2 | B Cells | 1.19305 | 3.657E−25 |
| *HLA-DQB1* | 1 | B Cells | 1.13693 | 1.5765E−21 |
| *IFIH1* | 2 | NK Cells | 0.659318 | 5.7183E−91 |
| *IFIH1* | 1 | NK Cells | 0.650195 | 1.0135E−72 |
| *IFIH1* | 3 | NK Cells | 1.26942 | 2.1223E−28 |
| *JAZF1* | 2 | CD8 T Cells | 0.660968 | 2.5189E−49 |
| *JAZF1* | 3 | B Cells | 1.19587 | 2.1541E−27 |
| *NFKB1* | 1 | B Cells | 0.853393 | 0 |
| *NFKB1* | 2 | Monocyte | 0.898239 | 2.013E−178 |
| *NFKB1* | 1 | CD4 T Cells | 0.898719 | 5.5342E−93 |
| *NFKB1* | 2 | B Cells | 0.92324 | 2.2148E−85 |
| *NFKB1* | 1 | Monocyte | 0.93036 | 2.4946E−47 |
| *NFKB1* | 3 | CD4 T Cells | 0.951296 | 1.1541E−22 |
| *NFKB1* | 2 | CD4 T Cells | 0.953377 | 1.9101E−21 |
| *NFKB1* | 1 | CD8 T Cells | 0.93407 | 1.2091E−20 |
| *NFKB1* | 2 | NK Cells | 0.93831 | 4.9184E−20 |
| *NR4A3* | 1 | B Cells | 0.767656 | 3.979E−110 |
| *NR4A3* | 2 | Monocyte | 0.888678 | 1.349E−24 |
| *RGS1* | 2 | Monocyte | 0.774416 | 3.3753E−67 |
| *RHOH* | 1 | B Cells | 0.878026 | 1.8915E−62 |
| *STAT4* | 3 | NK Cells | 1.13119 | 3.3902E−60 |
| *STAT4* | 2 | Monocyte | 0.787111 | 7.7766E−57 |
| *STAT4* | 3 | CD4 T Cells | 1.08251 | 1.9188E−24 |
| *STAT4* | 2 | B Cells | 0.848873 | 9.6623E−24 |
| *THEMIS* | 1 | CD8 T Cells | 0.800418 | 1.4367E−30 |
| *TNFAIP3* | 1 | Monocyte | 0.836954 | 2.064E−46 |
| *TNFAIP3* | 2 | Monocyte | 0.8774 | 1.9542E−36 |
| *TNFAIP3* | 1 | B Cells | 0.853863 | 4.5395E−34 |
| *TNFAIP3* | 2 | B Cells | 0.866305 | 1.06E−21 |
| *TOX* | 2 | CD8 T Cells | 0.803685 | 1.5961E−24 |

differences in proximity of disease loci did not reach significance for T1D SNV association, i.e., despite being associated with case status the differences were not driven by genetic disease association detectable in our sample size. However, the excess of early (T1) open chromatin traits in proximity of T1D loci suggests that studying genetic influences on immune regulation predisposing to T1D could require monitoring regulatory states even earlier in infancy.

While the study in TEDDY highlighted the need for longitudinal measures of the dynamic infant immune system and demonstrated that distinct immune signatures are present prior to T1D onset and could be predictive, the study lacked single-cell resolution that is added by our work. In both cases, identifying validated immune signatures that could stratify T1D disease risk will allow for early monitoring of infants at risk. Improved predictive value of biomarkers based on single nuclei in patients at risk could allow more targeted testing of interventional therapies to prevent T1D.

## Methods
### TRIGR study
The Trial to Reduce IDDM in the Genetically at Risk (TRIGR) tests the hypothesis that weaning to a hydrolyzed formula prevents β-cell autoimmunity and T1D in a randomized clinical trial. The primary outcome was negative. Clinical Trials.gov identifier: NCT00179777. TRIGR recruited a total of 2159 participants 2002–2007. The participants were followed up until the youngest child reached 10 years of age 2017. One hundred seventy-three children (8.0%) progressed to clinical T1D during follow-up to a median age of 11.5 years.

### Study participants
The TRIGR study is a randomized clinical trial testing the hypothesis that weaning to an extensively hydrolyzed formula prevents β-cell autoimmunity and T1D. The study recruited 2159 infants with HLA-conferred disease susceptibility [the high risk genotype *DQB1*02/ DQB1*03:02*, moderate risk genotypes *DQB1*03:02/x* (*x* not *DQB1*02, DQB1*03:01*, or *DQB1*06:02*), mild risk genotypes *DQA1*05-DQB1*02/y* (*y* not *DQA1*02:01-DQB1*02, DQB1*03:01, DQB1*06:02*, or *DQB1*06:03*) or the rare mild risk genotype *DQA1*03-DQB1*02/y* (*y* not *DQA1*02:01-DQB1*02, DQB1*03:01, DQB1*06:02*, or *DQB1*06:03*)] and a first-degree relative with type 1 diabetes from May 2002 to January 2007 in 78 study centers in 15 countries; 1081 were randomized to be weaned to a hydrolyzed casein formula and 1078 to a conventional formula. The follow-up of the TRIGR participants ended on February 28, 2017. β-cell autoimmunity and type 1 diabetes were primary outcomes, while secondary outcomes included age at T1D diagnosis and safety (adverse events). No differences were seen in the outcomes between the two arms. The intervention was safe, and no study-related serious events were observed in the trial[16]. All uses of human material have been approved by the Ethics Committee of the Hospital District of Helsinki and Uusimaa (Helsinki, Finland, HUS 617/E0/02). The guardians of all participating children gave their written informed consent. The participants attended the study centers at the age of 3, 6, 9,12, 18, and 24 months and subsequently annually until the youngest child turned 10. A sample for the isolation of PBMCs was collected at each visit and the sample from European participants was shipped overnight for isolation in the TRIGR Core Laboratory in Helsinki. The current study includes 98 European TRIGR participants, out of whom 49 progressed to clinical T1D during the follow-up (cases). Two-thirds of the participants in the current study are from Finland. The remaining children are from nine different European countries. Forty-nine autoantibody-negative control participants were matched with the cases for date of birth (±1 year) and geographical region. Seroconversion to autoantibody positivity was observed in the 49 cases at the mean age of 2.4 years (range 0.7–7.1 years), and T1D was diagnosed at the mean age of 6.6 years (range 1.0–13.7 years). The controls remained autoantibody negative throughout the follow-up. Each participant in the current

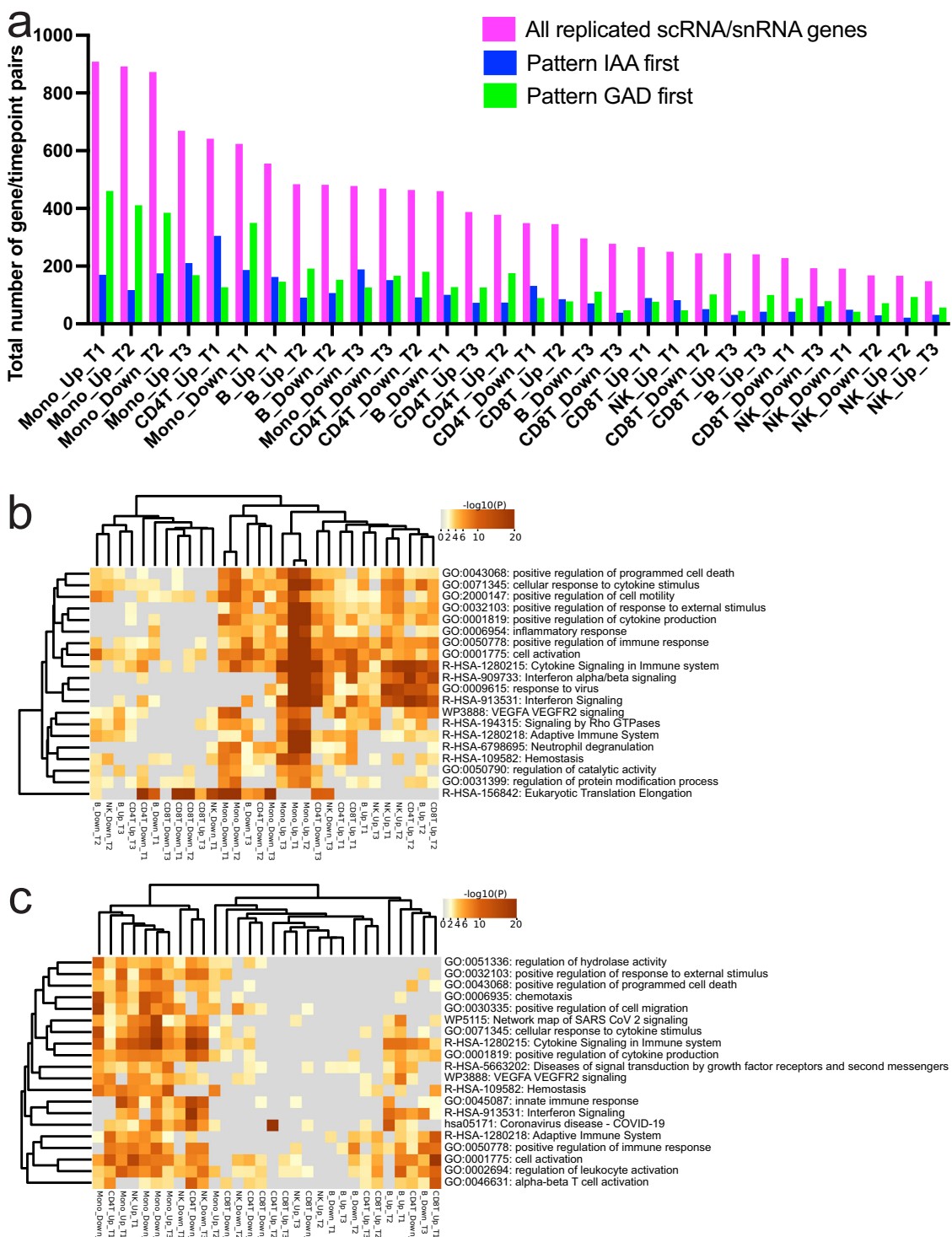

**Fig. 5 | Replication of case versus control differential expression patterns among case endophenotypes. a** Number of significant (y-axis) case vs. control signals (lavender bars) at each cell type / time point (x-axis) also significant in differential analyses of IAA first (blue bars) vs. GAD first (green bars). Early monocyte time points T1 / T2 patterns are showing most similarity to GAD first patterns. Proportionally CD8 T2 case patterns appear also to be originating more from GAD first subpopulation of cases. Source data are provided in the Source Data file. **b** Pathway analysis for GAD first vs. matched controls shows that large fraction of case activated pathways at early monocyte time points are predominanty active among GAD first endophenotype. Direction of effect (column labels) are based on GAD first expression as compared controls. Generated by Metascape (hypergeometric test p-values with Benhamini-Hochberg correction). **c** Pathway analysis for IAA first vs. matched controls is showing less remarkable enrichment in monocyte overexpressed pathways, but upregulated genes at first timepoint of CD4T cells with high overlap of overall case−control differences (a) points to pathways linked to adaptive immunity. Direction of effect (column labels) are based on IAA first expression as compared controls. Generated by Metascape (hypergeometric test $p$ values with Benhamini-Hochberg correction).

study contributed three samples; the first collected before the appearance of the first autoantibody (mean age 1.6 years, range 0.3–6.1 years), the second after the seroconversion to autoantibody positivity (mean age 3.0 years, range 0.6–8.2 years) and the third close to the diagnosis of T1D (mean age 6.8 years, range 1.0–13.6 years). The samples from the control participants were obtained at the corresponding ages. The detailed characteristics of the participants are presented in Table 1.

## Analysis of diabetes-associated autoantibodies

Diabetes-associated autoantibodies, i.e., insulin autoantibodies (IAA), GAD autoantibodies (GADA), and autoantibodies to islet antigen 2 (IA-2A) and zinc transporter 8 (ZnT8A), were quantified with the use of specific radiobinding assays in the Scientific Laboratory, Children's Hospital, University of Helsinki, Helsinki, Finland, in all samples obtained from the TRIGR participants during their follow-up[31]. The cut-off for autoantibody positivity was 1.57 relative units (RU) for IAA, 5.36 RU for GADA, 0.78 RU for IA-2A, and 0.50 RU for ZnT8A based on samples from 370 non-diabetic children. The laboratory has taken part in the international Islet Autoantibody Standardization Program on a regular basis. In 2023, the sensitivities of the IAA, GADA, IA-2A, and ZnT8A radiobinding assays were 60%, 82%, 74% and 74%, respectively, and the corresponding specificities were 100%, 99%,100% and 98%.

## Endpoints

The primary endpoint was progression to overt T1D and included comparison between the cases and controls at the three time points analyzed. Secondary analyses comprised a comparison of the "IAA first" and "GADA first" endotypes among the cases. The strict criterion for "IAA first" marked that IAA was the only detectable autoantibody in the first autoantibody-positive sample, while the looser criterion indicated that the first autoantibody-positive sample tested positive for multiple autoantibodies, including IAA but not GADA. Similarly, the strict criterion for "GADA first" marked that GADA was the only detectable autoantibody in the first autoantibody-positive sample, while the looser criterion indicated that the first autoantibody-positive sample tested positive for multiple autoantibodies, including GADA but not IAA. IAA and GADA are the two most common autoantibody specificities that appear during the disease process leading to clinical T1D. Both IA-2A and ZnT8A are rare ( < 5%) as the first appearing autoantibody. The sampling interval of 3-12 months may be too long to allow the identification of the single first-appearing autoantibody in all cases. The most conspicuous differences between endotypes have been observed between the IAA first and GADA first endotypes even when using the loose definition of the endotype. For endophenotype analyses we utilized strict criterion "IAA first" ($n = 21$) and looser "GADA first" ($n = 11$) definitions and generated differential expression / peak analyses using the paired controls against each major cell lineage and timepoint. SnRNA signals replicated at qv = 0.1 in min of two data layers were kept for comparison to overall case – control analyses to explain the endophenotypes' gene network contributions of to overall case – control differences.

## Isolation of PBMCs from fresh blood samples

PBMCs were separated from fresh heparinized venous blood samples by Ficoll (GE Healthcare, Uppsala, Sweden) isogradient centrifugation. Prior to the isolation of PBMCs, blood plasma was separated with centrifugation and the separated plasma was substituted by an equal volume of phosphate buffered saline (PBS). The diluted blood was then placed on top of 5 mL of Ficoll in 15 mL conical Falcon tubes. The tubes were centrifuged for 25 minutes at 800 x $g$ at room temperature (RT) with no brake during the deceleration step. The upper plasma layer was removed by aspirating with a sterile individually packed plastic Pasteur pipette. Then, the white layer containing the mononuclear cells was collected and directly transferred into a new 15 mL Falcon

tube containing 10 mL PBS. The collected cells were washed three times with PBS. After the washing steps, the supernatant was carefully removed and the cell pellet was suspended in RPMI 1640 cell culture medium (Thermo Fisher Scientific, Cat. No. 11875093) supplemented with 5% of heat inactivated human AB serum (Innovative Research, Cat. No. ISERABHI100ML), 2 mM L-glutamine (Sigma-Aldrich, Cat. No, G8540) and 25 µg/mL gentamicin (Sigma-Aldrich, Cat. No. G1397). The cell density and viability was then analyzed using a hematocytometric chamber (Burker) and Trypan blue staining (Thermo Fisher Scientific, Cat. No. 15250061).

## Freezing the isolated PBMCs

Isolated PBMCs were transferred into a new conical Falcon tube and the cells were centrifuged at room temperature (RT) for 5 minutes at $400 \times g$ with centrifuge brake on. The supernatant was pipetted off and the cells were resuspended in cold ( 4 °C) RPMI 1640 with 5% of heat inactivated human AB serum, 2mM L-glutamine and gentamicin (25 ug/mL). At this point the cell density was between $4–16 \times 10^6$/mL. Then an equal volume of cold ( + 4 °C) RPMI 1640 cell culture medium with 5% of heat inactivated human AB serum, 2 mM L-glutamine, gentamicin (25 ug/mL) and 20% of dimethyl sulphoxide (DMSO) (Sigma-Aldrich, Cat. No. D2650) was added dropwise slowly to the cell suspension while swaying the tube gently. Cell suspension was then aliquoted into the cryovials. One mL of the cell suspension was transferred to cryovial tubes. The tubes were closed and put immediately in a MrFrosty container (Thermo Fisher Scientific, Cat. No. 5100-0001) with iso-propyl alcohol for freezing the cells in a temperature-controlled way. The MrFrosty container with the cells was then immediately transferred into a −80 °C freezer overnight. The cryovials were transferred the next day into the automated gas phase liquid nitrogen freezer for storing at −180 °C until thawed and analyzed.

## Batching the thawed PBMCs

Cell samples were thawed in a group of 5 samples at a time in a 37 °C water bath until a small ice crystal remained in the cryovials. 1 mL of pre-warmed thawing medium (RPMI 1640 culture medium containing 2 mM L-glutamine, 25 mM/L HEPES (Thermo Fisher Scientific, Cat. No. 15630080), 25 µg/mL gentamycin and 10% of heat-inactivated AB serum), were added to the cryovials slowly. The cell suspensions were transferred to the 15 mL conical tubes containing 10 mL of thawing medium. The cells were centrifuged at 300 x $g$ for 8 minutes, and the cell supernatant was discarded. The cell pellet was gently resuspended in a volume of 0.5 mL of room-temperature Thawing Medium. An evaluated volume of cell suspension containing approximately $1 \times 10^6$ cells per sample of 42–46 cell samples were pooled together. After filtering the pooled cell sample with a 40 µm cell strainer, the cells were counted, and the viability was assessed. On average, the viability of the cells was 89% with a standard deviation of 9.1%. After viability analysis, the cell suspension was divided between two 15 ml conical tubes and centrifuged at 300 x $g$ for 8 minutes at 4 °C. The supernatant was discarded, and the cell pellets were resuspended in 1 mL of cold DMSO-containing cryopreservation medium and transferred to cryovials. The cryovials were placed into a slow-freeze cryopreservation chamber in a −80 °C freezer. On the next day the frozen cells samples were relocated into a liquid nitrogen storage. The frozen cryovials were then transported from Helsinki to Kansas City in a cryoshipper.

## DNA genotype analyses and genotype demultiplexing

The high-throughput microarray genotyping data (from Axiom when available) was merged with variant calls from WGS samples (TRIGR_1045832, TRIGR_1061181, TRIGR_1062514, TRIGR_1062558, TRIGR_1071138, TRIGR_1085671). The donor-multiplexed multiome data was deconvoluted using demuxlet v2[32,33]. Reads for each individual cell were identified by the unique cell barcode (CID), and the genotypes observed on the cell-specific reads at shared, highly variable

SNP positions ($n = 436683$ shared, variable SNPs) were compared against the merged imputed and WGS genotype matrix of the pool to accurately assign each cell barcode (CID) to an individual donor based on the distinctive combination of variants observed. Ambiguous cases such as cell doublets were removed from the data.

## Single-cell RNA, single nuclei RNA and ATAC sequencing

Thawed cells were centrifuged at $300 \times g$ for 8 minutes and resuspended in 10 mL of Thawing Medium consisting of IMDM (ATCC, Cat. No. 30-2005) supplemented with 10% fetal bovine serum (Fisher Scientific, A5669801), 100 Units/mL Penicillin, and 100 µg/mL of Streptomycin (Thermo Fisher Scientific, Cat. No. 15140122). Cells were washed once more in Thawing Medium then twice in PBS + 0.04% BSA and filtered through a 40-µm nylon mesh cell strainer (Fisher Scientific, Cat. No. 07201430). A Countess II automated cell counter (Thermo Fisher Scientific, Cat. No. AMQAF1000) was used to assess the cell count and viability of the pool. Approximately 25,000 cells were loaded into each of four wells of a 10x Genomics Chromium Chip B, and single-cell RNA (scRNA)-seq was performed using 10x Genomics Chromium Single Cell 3' Reagent Kits v3 (Cat. No. PN-1000121) according to the manufacturer's protocol. Single nuclei ATAC (snATAC) was then performed in parallel with scRNA. Nuclei were isolated from an aliquot of the pooled cells following the 10x Genomics Demonstrated Protocol: Nuclei Isolation for Single Cell ATAC Sequencing with an optimized lysis time of 3 min. Approximately 15,300 transposed nuclei were loaded into each of four wells of a 10x Genomics Chromium Chip E, and snATAC-seq was performed using 10x Genomics Chromium Single Cell ATAC Reagent Kits v1 (Cat. No. PN-1000175) according to the manufacturer's protocol. All scRNA and snATAC libraries were loaded on an Illumina NovaSeq 6000 sequencer (Illumina) using S4 Reagent Kit v1.5 (200 cycles) and sequenced to the recommended depths.

10X Genomics Chromium Single Cell Multiome ATAC + Gene Expression protocol was deployed in independent replicate cell pools for simultaneously assessing nuclear RNA expression and open chromatin by ATAC profiles on the Chromium Instrument with Next GEM technology. Pooled cells were thawed and washed as described above. Nuclei were isolated from pooled PBMCs according to 10x Genomics Demonstrated Protocol CG000365 using an input of 1 million cells and a lysis time of 3 minutes. Nuclei were stained with trypan blue and quantified using a Countess 3 automated cell counter. For each pool, approximately 16,100 transposed nuclei were loaded into each of eight wells of a 10x Genomics Chromium Next GEM Chip J to target at least 1000 nuclei per donor retained after data filtering for cell quality and doublets using conservative estimates (see rationale for this strategy in more details below). Simultaneous single nuclei RNA (snRNA)/ATAC-seq was performed using 10x Genomics Chromium Next GEM Single Cell Multiome ATAC + Gene Expression Reagent Kits (Cat. No. PN-1000283) according to the manufacturer's protocol. All Multiome libraries were loaded on an Illumina NovaSeq 6000 sequencer using S4 Reagent Kit v1.5 (200 cycles) and sequenced to the minimum depth of 50,000 GEX and 25,000 ATAC reads per nuclei, respectively.

## Processing and differential gene / peak expression in scRNA/snATAC and snRNA-seq analyses

Primary analysis of NovaSeq6000 genome sequences, to produce sequence reads, was performed with Illumina Real Time Analysis (RTA) software and bcl2Fastq2-20. Single cell RNA, snRNA and ATAC 10X Genomics data was processed by cellranger-4.0.0 and cellranger-atac-1.2.0. The Cell Ranger Single Cell Software was used to perform sample demultiplexing, barcode processing, and single cell 3' gene counting for RNA data, along with fragment-mapping and counting for ATAC data. The R-based Seurat 4.0 package[34] was used for QC, analysis, and exploration of the data.

## Processing and observed/expected read depths in single-nuclei Multiome augmented with snATAC data

Seurat 4.0 package was used for QC, analysis, and exploration of the scRNA-seq and Multiome data and to identify and interpret sources of heterogeneity from the measurements. Quality filtering was based on the number of unique molecular identifiers (UMI) and/or mitochondrial content in each cell. For snATAC, we applied Signac (https://CRAN.R-project.org/package=Signac) workflow[35], which is based on term frequency-inverse document frequency (TF-IDF) normalization and dimensionality reduction using singular value decomposition (SVD). Quality filtering was based on number/fraction of reads in peaks, transcriptional start site (TSS) enrichment score, ratio of reads in blacklist (non-unique genomic segments contaminating chromatin datasets). ATAC signal was evaluated in peaks taken from lymphoid and myeloid/erythroid DNase Hypersensitivity Sites (DHS)[36] showing 34,555/99,779 snATAC peaks overlapping lymphoid, and 22,707/99,779 peaks overlapping myeloid DHS (total overlaps 51,208/99,779).

We used unsupervised clustering and discovery of cell types/states and dimensionality reduction by the Uniform Manifold Approximation and Projection (UMAP) method as implemented in Seurat[19]. We manually curated all cell annotations provided by Seurat based on supervised literature mining. For Multiome datasets, the datasets were normalized using SCTransform and then integrated, using Seurat v4. Information from both scRNA and scATAC modalities were integrated independently for each cell using a weighted-nearest neighbor (WNN) approach. This incorporates information from both RNA and ATAC data robustly, without regard to the relative quality of each dataset.

Differential analysis between groups (e.g. case versus control) for RNA genes and ATAC peaks was performed by merging the cell types into five main groups guided by Seurat cell label transfers where "CD4", "CD8", "NK" and "B-cell" included all cells in any cluster with lineage label (e.g. "NK" included nuclei from NKT and NK CD56bright/dim labeled clusters), "Monocyte" included all Monocyte (e.g. CD14, CD16) and DC labeled nuclei. Merging the cell types ensured that each of the merged cell types was well-represented by over three quarters of the individuals in the study at each time point, ensuring the generality of the results and making maximal utility of the sequencing performed. Few smaller cell populations where omitted ("not assigned") and we verified no differential representation of these cell populations among cases and controls (Supplementary Figs. 1–3). Following collapse of sublineage clusters we used data of nuclei from five parent lineage clusters and applied Fisher's exact test to compare the total number of reads in a particular peak/gene ("observed") and the total number of reads in all other peaks/genes ("expected") for one group and comparing to the observed/expected total reads for the second group.

In both the snRNA dataset and the integrated snATAC, we computed correlation of RNA and ATAC signals near 141 known T1D-related single nucleotide polymorphism (SNP) loci. We extracted the imputed genotype value for each sample at the SNP, represented as an alt allele count of 0, 1, or 2 for genotypes 0/0, 0/1, and 1/1, respectively. SCT normalization was used for RNA merging, macs2peaks were used for the ATAC, using the Seurat and Signac packages. The read counts were normalized per 1 M total gene expression or ATAC reads for each individual and time point. For each of the five parent lineage clusters, we looked at genes (for the multiome dataset) or peaks (for the ATAC dataset) within 1Mbp of a SNP locus and used Spearman rank correlation to compute the rank correlation coefficient rho and P-value of the normalized signal in a particular peak/gene for each individual – total number of reads in a particular peak/gene divided by the total number of reads in all peaks/genes – against the matched genotype value of the SNP for each individual. As with other analyses we utilized five aggregated cell types CD4T, CD8T, B-cell, NK-cell and monocytes. Multiple testing was accounted for by Bonferroni correction using the number of tests applied per cell type and data modality. To investigate

age-dependent gene expression, the linear regression model (lm) in the R statistics package was used to perform a linear regression analysis, with the "Age at Draw" for each sample used as predictor and the pseudobulk RNA expression as the response variable, and multiple testing accounted for by FDR from the p.adjust package.

## Statistics and reproducibility

Sample size was the maximum number of samples from the longitudinal TRIGR cohort study, where there was a matching T1D case and control sample with data available collected at multiple timepoints. Similarly, the outcomes of disease progression and subtype determination determined which subtype groupings were available to be tested.

We excluded from our sample selection from the TRIGR cohort those cases that could not be matched to an appropriate control samples (as well as vice-versa). In the single-cell data, cells which were not unambiguously associated with a single genotype were excluded, as well as cells with insufficient number of unique UMIs or exceptionally high mitochondrial-associated reads. We used Fisher Exact testing for differential expression/accessibility and filtered the results to require a minimum number of cells and samples for testing. The TRIGR study was an international, randomized, double-blinded trial testing whether hydrolyzed infant formula compared to cow's milk-based formula decreased risk of developing T1D based on a child's genetic susceptibility[17]. The single cell experiments involved pooled cells from multiple case and control individuals which were randomly sequenced. Within our experiments, we also evaluated the reproducibility of our results by comparing results between data layers and across different q-values.

## Reporting summary

Further information on research design is available in the Nature Portfolio Reporting Summary linked to this article.

## Data availability

Data are stored in the European Genome-phenome Archive (EGA, https://ega-archive.org) under accession code EGAD50000001257. Individual participant data are shared in a de-identified format to protect the identity of the participants. There are no restrictions on who the data can be made available to or for what purpose. Please contact Marja Salonen (marja.salonen@helsinki.fi) to request access. One to six working days is the expected time frame for a response to access requests. The data will be available for 12 months once access has been granted. All data are included in the Supplementary Information or available from the authors, as are unique reagents used in this Article. The raw numbers for charts and graphs are available in the Source Data file whenever possible and are provided with this paper. Source data are provided with this paper.

## Code availability

Code used in the analysis are available on GitHub at https://github.com/ChildrensMercyResearchInstitute/trigr.

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

## Acknowledgements

T.P. holds a Fred and Dee Lyons Endowed Chair in Pediatric Genomic Medicine. Grant support for the study was from Academy of Finland (grant 350455) to M.K. The authors thank the TRIGR Study Group for making the PBMC samples available from the TRIGR children analyzed in this study. The TRIGR study was supported by the Eunice Kennedy Shriver National Institute of Child Health and Human Development (NICHD) and National Institute of Diabetes and Digestive and Kidney Diseases, National Institutes of Health (grants HD040364, HD042444 and HD051997), Canadian Institutes of Health Research, JDRF and the Commission of the European Communities (specific RTD program "Quality of Life and management of Living Resources", contract number QLK1-2002-00372 "Diabetes Prevention") and the EFSD/JDRF/Novo Nordisk Focused Research Grant.

## Author contributions

T.P. and M.K. designed the study, acquired data, analyzed data, and wrote the manuscript. E.G. and T.B. designed and supervised experiments. J.H., A.V., O.V., and M.K. were responsible for sample collection, sample storage, and the clinical information for the children. W.C., J.J.J., B.Y., and S.K. were responsible for bioinformatic analysis. R.M. assisted with manuscript and figure preparation. J.I. was responsible for the HLA genotyping of the participants. J.P.K. was the PI for the TRIGR Data Management Unit. All authors reviewed/edited the manuscript.

## Competing interests

The authors declare no competing interests.
