## [Transparent Peer Review file · Nature Communications]

Evolving epigenomics of immune cells at single nuclei resolution in children en route to type 1 diabetes

Corresponding Author: Dr Tomi Pastinen

Version 0:

Reviewer comments:

Reviewer #1

(Remarks to the Author)

I appreciated the opportunity to review this manuscript from Pastinen et al, which provides novel data across the spectrum of seroconversion to T1D development. A few changes would strengthen my enthusiasm for this overall nice work.

Major:

Figures 4& 5 illustrate pathways of interest by cell type, but the data are not very clearly presented for endotypes or for case/control analyses. How variable were these pathways between individuals? Were all pathways equally represented in all individual participants? More figures throughout that get to the variability between individuals in terms of both technical variation and immune response would be welcome – indeed, the only figure in the main text presenting individual-level data is figure 1b, which really highlights the importance of understanding between-individual variation in the dataset. In particular, individual level data to illustrate the difference between GAD-first and INS-first endotypes is critical to understanding the strength of this finding, especially given the relatively low N for each strict endotype. I think Figure 3 could be cut or moved to supplemental if figure count is a concern.

Response to previous reviews indicates that there was no evidence of a difference between cases and controls by trajectory analyses – this should be included in the manuscript as it is a natural question readers will ask, and some discussion of why there was no difference would be welcome.

Response to previous reviews also notes the viability impacts of multiple freeze thaws – but the authors did not comment on whether there was a differential effect by donor or cell type. Monocytes in particular may have been lost to freeze thaw.

If allowable by editors, references to prior work should be increased throughout the discussion. In particular the discussion mentions prior studies of BACH2 and TNFAIP3 as examples but does not cite the prior literature, and there are other findings around interferon signatures in T1D risk that would be relevant to cite.

Minor:

Line 66 – “These insights relied on a unique study design within TRIGR...” – it is not clear which insights are referred to. Overall diabetes risk and understanding of endotypes and heterogeneity of T1D development have come from a variety of studies worldwide. Perhaps a section of the introduction was unintentionally cut, as TRIGR is not clearly defined either.

Figure 1a states that the 98 individuals are in HLA-risk matched case/control pairs, but Table 1 shows that the n for both high (DQB1*02/DQB1*03:02) and moderate (DQB1*03:02/x) risk HLAs is higher for cases than controls, and the n for mild and rare mild HLA (as defined in methods) is much lower. The methods only mentions matching on age and geographic region. Figure 1a should be corrected along with the methods if any other matching characteristics were used beyond age and geography.

Figure 1e-g would be more useful with a legend annotating the cell clusters.

N's need to be more clearly displayed and provided in each legend – how many individuals dropped out for QC reasons from any given figure, how many individuals are in the GAD v INS first endotype, etc. Does the number of cells analyzed vary between figures or does Supplemental Table 1 represent the values used for every analysis?

Supplemental Figures 4 c-f are helpful to understand the relative composition of the pools, and should include individual points per sample rather than solely box and plunger plots. Supplemental figure 4 e-f are not described in the associated figure legend.

Table 1 includes age at end of followup but not age at sampling timepoints; adding this could perhaps provide information on why there is higher differential expression early in the study than later.

Reviewer #2

(Remarks to the Author)

I reviewed this work at Nature Medicine and again on transfer to Nature Communications. The authors have satisfactorily addressed my prior comments (and those from other reviewers), and I have no further technical or interpretive concerns.

Reviewer #3

(Remarks to the Author)

I acknowledge the difficulty of performing the proposed flow cytometry validation experiments in a controlled, adequately matched sample set before autoantibody seroconversion, and that the concordant gene modulation patterns observed across single-gene platforms and "within cohort" replication approach add validity to the conclusions.

Version 1:

Reviewer comments:

Reviewer #1

(Remarks to the Author)

No further concerns identified.

(Remarks on code availability)

I have not reviewed the code.

We thank the reviewers for their time and consideration in reviewing our manuscript. All remarks have been addressed. Below is a point-by-point response to each reviewer comment.

REVIEWER COMMENTS

Reviewer #1 (Remarks to the Author):

I appreciated the opportunity to review this manuscript from Pastinen et al, which provides novel data across the spectrum of seroconversion to T1D development. A few changes would strengthen my enthusiasm for this overall nice work.

Major:

Figures 4& 5 illustrate pathways of interest by cell type, but the data are not very clearly presented for endotypes or for case/control analyses. How variable were these pathways between individuals? Were all pathways equally represented in all individual participants? More figures throughout that get to the variability between individuals in terms of both technical variation and immune response would be welcome – indeed, the only figure in the main text presenting individual-level data is figure 1b, which really highlights the importance of understanding between-individual variation in the dataset. In particular, individual level data to illustrate the difference between GAD-first and INS-first endotypes is critical to understanding the strength of this finding, especially given the relatively low N for each strict endotype. I think Figure 3 could be cut or moved to supplemental if figure count is a concern.

Response- We appreciate the comments of variability across individuals. We acknowledged this complexity from the design of the study being the largest single cell, longitudinal study ever done with individuals at risk of a complex disease and matched controls with same risk profile. It is not unexpected that “low hanging fruit”, i.e. simple classifier was not uncovered. There is substantial variation in individual molecular features showing association with T1D development even in this highly matched set of children. However, we believe our data starts to show clues where we should look for new triggers of disease to develop biomarkers and interventions in individuals at very high risk of T1D. To make sure that extent and ability of our data in interpreting risk for T1D today is not overstated we show full raw data in the Figure 3, new panels (a) showing raw data on some of the separately discussed features (genes) with relevance to T1D vary in cases and controls (highlighted separately in Table 3). The variability motivated the approach we chose for overall discussion of results – we emphasize the need for repeated observation of same feature in more than one data layer and therefore feel that original Figure 3 panels should be preserved. We have now also added the results of enrichment analyses across data layers showing that while individual features (genes or peaks) can be highly variable they are replicated independently at high rates. We also note that pathway enrichments even for independently replicated genes are variable across individuals (Figure 4). Finally, we have clarified the approach taken in subgroup (endotype) analyses linked to results in Figure 5. As the reviewer points out there are uncertainties with smaller

sample, consequently we did not pursue pure discovery but rather queried if the observed case – control differences (Figures 3 and 4) could predominantly originate from either endotype. We have clarified this approach and added raw data similar to new Figure 3a as a Supplementary Figure 8.

Lines 366-367- “Individual case – control associations feature associations show high variability (**Figure 3a**).”

Lines 484-486- “We focused on features replicated in full case – control analyses (**Figure 3**) given the smaller sample subpopulations and contrasted features that showed association in one endotype predominantly (**Supplementary Figure 8**).”

Response to previous reviews indicates that there was no evidence of a difference between cases and controls by trajectory analyses – this should be included in the manuscript as it is a natural question readers will ask, and some discussion of why there was no difference would be welcome.

Response- We have now included an additional result at the end of paragraph “Differential multiomics immune signatures among T1D cases vs control “ and included new Supplementary Table 13 and description of linear model testing in ONLINE METHODS showing that robust age dependent changes in gene expression (snRNA) are shared between cases and controls.

Lines 298-301- “To investigate age-dependent gene expression, the linear regression model (lm) in the R statistics package was used to perform a linear regression analysis, with the “Age at Draw” for each sample used as predictor and the pseudobulk RNA expression as the response variable, and multiple testing accounted for by FDR from the p.adjust package.”

Lines 375-389- “Across all data layers the level of replication is more than expected by chance. For example, in the case-control differential expression, we expect a significant number of genes observed in T1 to be replicated in T2, and similarly between T2 and T3. We observe non-random enrichment between observed T1/T2 and T2/T3 overlapping genes/peaks compared to the expected overlap even at lenient P-value thresholds, which grows stronger as the stringency of the significance is increased: at $q < 0.5$ we see a small enrichment (snRNA 2-2.6X, ATAC 1.6X-4X, scRNA 1.7-3.3X), but as we increase to $q < 0.1$ we observe a substantial enrichment (snRNA 4.8X-7.4X, ATAC 5.2X-507X, scRNA 3.6X-11X).

We also queried differences in age-dependent expression by a linear expression separately in cases and controls across cell lineages (**Supplementary Table 13**) to understand if nuclear epigenome traits developed differently among groups (q-value cut-off 0.1). The age-dependent snRNA traits that were shared in both case and control groups were substantially more robust than group specific traits showing median q-value of 0.001 (median $r^2 = 0.11$) as compared to median q-value of 0.03 (median $r^2 = 0.06$). Over 80% of top (1%:tile) age-dependent associated traits were shared in cases and controls underscoring that majority of expression trajectories are independent of case status.”

Response to previous reviews also notes the viability impacts of multiple freeze thaws –

but the authors did not comment on whether there was a differential effect by donor or cell type. Monocytes in particular may have been lost to freeze thaw.

Response- This is true, we expect biological differences in viability of cell populations over time. Post-hoc this is not feasible to assess for cell populations without cell sorting over time. However, the cases and controls were collected and stored by the TRIGR trial in parallel. We then performed pooling at the same time on stored specimens and freeze/thaw cycles are identical across the case and control populations. Also, the age-matching should reduce differences in differential primary storage times between cases and controls. Altogether, considering the study design we believe the freeze / thaw cycles do not play a role in the presented results.

If allowable by editors, references to prior work should be increased throughout the discussion. The discussion mentions prior studies of BACH2 and TNFAIP3 as examples but does not cite the prior literature, and there are other findings around interferon signatures in T1D risk that would be relevant to cite.

Response- We have added references to the discussion, line 545.

Minor:

Line 66 – “These insights relied on a unique study design within TRIGR...” – it is not clear which insights are referred to. Overall diabetes risk and understanding of endotypes and heterogeneity of T1D development have come from a variety of studies worldwide. Perhaps a section of the introduction was unintentionally cut, as TRIGR is not clearly defined either.

Response - We have now defined the TRIGR study in the introduction by transferring two sentences from the first paragraph in the ONLINE METHODS. The definition of endotypes requires a tight sampling schedule in early life. The TRIGR protocol comprised such a schedule with sampling every third month including also PBMCs during the first year of life.

Lines 71-73- “TRIGR was a randomized clinical trial designed to assess whether it is possible to prevent β -cell autoimmunity and clinical T1D by weaning high-risk infants to an extensively hydrolyzed formula. The primary outcomes of the trial turned out to be negative”

Lines 85-86- “TRIGR recruited altogether 2159 participants 2002-2007. The participants were followed up until the youngest child reached 10 years of age 2017”

Figure 1a states that the 98 individuals are in HLA-risk matched case/control pairs, but Table 1 shows that the n for both high (DQB1*02/DQB1*03:02) and moderate (DQB1*03:02/x) risk HLAs is higher for cases than controls, and the n for mild and rare mild HLA (as defined in methods) is much lower. The methods only mentions matching on age and geographic region. Figure 1a should be corrected along with the methods if any other matching characteristics were used beyond age and geography.

Response- No matching characteristics other than age and geography were used. Figure 1a has been revised accordingly.

Figure 1e-g would be more useful with a legend annotating the cell clusters.

Response- Cell cluster legend has been added to Figure 1e that applies to e-g.

N's need to be more clearly displayed and provided in each legend – how many individuals dropped out for QC reasons from any given figure, how many individuals are in the GAD v INS first endotype, etc. Does the number of cells analyzed vary between figures or does Supplemental Table 1 represent the values used for every analysis?

Response- We have now added subtype to Supplementary Table 1 and summarized the number of individuals and total cells for each condition in Supplementary Table 2 to clarify the number of individuals and cells involved in the analysis.

As seen in Supplementary Table 2, most individuals are present in most datasets at most timepoints – in case and controls the number of individuals present range from 45-49, for GADA and GADA controls 10-11 and for IAA and IAA controls 20-21 individuals, with the specific individuals present in each dataset at each timepoint and their cell contributions detailed in Supplementary Table 1.

Lines 318-321- “A total of 307,744 case and 330,006 control cells or nuclei passed the QC, including a total of 85628 looser GADA and 125789 strict IAA cells or nuclei and total of 82487 and 136331 matching control cells or nuclei (summarized counts of individuals and cells by dataset, timepoint and condition are in **Supplementary Table 2**).”

Supplemental Figures 4 c-f are helpful to understand the relative composition of the pools, and should include individual points per sample rather than solely box and plunger plots. Supplemental figure 4 e-f are not described in the associated figure legend.

Response- Supplemental Figures 4c-f have been updated to include individual plots. We thank the reviewer for catching the figure legend issue and have updated it.

Table 1 includes age at end of followup but not age at sampling timepoints; adding this could perhaps provide information on why there is higher differential expression early in the study than later.

Response- Age at the three sampling time points has been added to Table 1.

Reviewer #2 (Remarks to the Author):

I reviewed this work at Nature Medicine and again on transfer to Nature Communications. The authors have satisfactorily addressed my prior comments (and those from other reviewers), and I have no further technical or interpretive concerns.

Response- We thank the reviewer for their contribution to this manuscript.

Reviewer #3 (Remarks to the Author):

I acknowledge the difficulty of performing the proposed flow cytometry validation experiments in a controlled, adequately matched sample set before autoantibody seroconversion, and that the concordant gene modulation patterns observed across single-gene platforms and "within cohort" replication approach add validity to the conclusions.

Response- We thank the reviewer for their contribution to this manuscript.